# Perioperative Profiling of a Disintegrin and Metalloprotease with Thrombospondin Type 1 Motif, Member 13 (ADAMTS13) Activity in Cardiac Surgery: Kinetics and Mechanistic Insights

**DOI:** 10.3390/jcm14144936

**Published:** 2025-07-11

**Authors:** Bernhard Strasser, Johann Knotzer, Selina Sartori, Bernhard Poidinger, Oskar Kotzinger, Christian Irsara, Gerald Lirk, Carolin Gunz, Alexander Haushofer

**Affiliations:** 1Institute of Clinical Chemistry and Laboratory Medicine, Klinikum Wels-Grieskirchen, 4600 Wels, Austria; 2Department of Anaesthesiology and Critical Care, Klinikum Wels-Grieskirchen, 4600 Wels, Austria; johann.knotzer@klinikum-wegr.at (J.K.);; 3Central Institute of Clinical and Chemical Laboratory Diagnostics, University Hospital Innsbruck, 6020 Innsbruck, Austria; 4Research Center Hagenberg, University of Applied Science Upper Austria, 4232 Hagenberg im Mühlkreis, Austria; 5Landesklinikum Hainburg, 2410 Hainburg an der Donau, Austria

**Keywords:** ADAMTS13 activity, von Willebrand factor, VWF, cardiac surgery, perioperative hemostasis

## Abstract

**Background:** The enzyme A Disintegrin and metalloprotease with thrombospondin type 1 motif, member 13 (ADAMTS13) regulates hemostasis by cleaving von Willebrand factor (VWF) multimers. ADAMTS13–VWF axis dysregulation leads to different thrombotic conditions. This study investigated changes in ADAMTS13 activity during major cardiac procedures and their relationship to VWF changes and clinical complications. **Methods:** A total of 628 ADAMTS13 activity and inhibitor measurements were carried out in 168 patients who underwent cardiac surgery. ADAMTS13 activity was measured after the initiation of anesthesia and daily for up to 6 days postoperatively via Technozym chromogenic ELISA. The von Willebrand factor antigen (VWF:Ag) and collagen binding (VWF:CB) were also measured. Clinical complications and correlations with liver function biomarkers were also assessed. **Results:** ADAMTS13 activity significantly decreased during surgery, with mean values markedly decreasing from preoperative to postoperative measurements (*p* = 0.01). A clear inverse relationship between ADAMTS13 activity and the VWF:CB/VWF:AG ratio was observed, indicating that increased high-molecular-weight VWF multimers are associated with decreased ADAMTS13 activity. Correlation analyses (CHE, Spearman’s rho = 0.39) indicated that the reduction in ADAMTS13 activity was not attributable to impaired liver synthesis but likely resulted from peripheral consumption, potentially influenced by surgical stress. **Conclusions:** Perioperative reductions in ADAMTS13 activity are associated with an accumulation of high-molecular-weight VWF multimers and a higher incidence of postoperative complications. These results demonstrate that ADAMTS13 could be a useful perioperative risk biomarker for cardiac surgery patients.

## 1. Introduction

The protease Disintegrin and metalloprotease and thrombospondin type 1 motif 13 (ADAMTS13) is important in the blood coagulation process [1,2]. Hepatic stellate cells are the main site of ADAMTS13 production. ADAMTS13 remains inactive in the bloodstream until it encounters VWF, particularly under the high shear conditions characteristic of small arteries. The active form of ADAMTS13 terminates the accumulation of ultralarge VWF multimers through VWF cleavage between tyrosine 1605 and methionine 1606, thus preventing uncontrolled platelet aggregation or bleeding events. ADAMTS13 has a half-life of approximately two to three days in human individuals [2,3,4]. The heart–lung machines (HLMs) used in cardiac surgery can have substantial effects on human blood clotting. Coagulation system activation occurs when blood comes into contact with the artificial surfaces of the HLM, resulting in consumption coagulopathy that depletes coagulation factors and platelets. The mechanical forces on platelets impair function, leading to decreased blood coagulation ability. The combination of surgical trauma and these conditions leads to impaired postoperative coagulation homeostasis, which elevates the risk of bleeding complications in cardiac surgery patients [5].

Coagulation disorders are influenced not only by surgical interventions but also by underlying cardiovascular conditions such as aortic stenosis. The narrowed heart valve in aortic stenosis produces substantial blood shear stress because of its restricted opening. The mechanical stress caused by this condition modifies VWF structure by damaging its high-molecular-weight multimers, which play a vital role in blood clotting. The enzyme ADAMTS13 shows increased activity toward large VWF multimers during these conditions, leading to their breakdown. The diminished VWF levels reduce its ability to help platelets bind to injured blood vessels, thus leading to higher bleeding risks. The main manifestation of Heyde’s syndrome, which links aortic stenosis to bleeding symptoms and vascular malformations, frequently occurs in the gastrointestinal tract and presents as angiodysplasia-related bleeding. The normalization of shear forces following aortic valve replacement through surgical or transcatheter procedures leads to the recovery of high-molecular-weight VWF multimers. The procedure triggers a rapid recovery process that starts within hours to days and leads to substantial improvement of bleeding symptoms. The measurement of these multimers serves as a diagnostic tool to evaluate the severity of aortic stenosis and to distinguish between severe and pseudo-severe cases, especially when patients have low-flow or low-gradient conditions [6].

At present, multiple established test systems exist for examining ADAMTS13 activity. Enzyme-linked immunosorbent assays (ELISAs) are leading analytical tools because they exhibit good performance and exact measurement capabilities [7]. The runtimes of these ELISA methods are four to five hours. Fluorescence resonance energy transfer (FRET)-based assays utilize recombinant substrates. The performance levels of ELISA and FRET assays are equivalent in terms of sensitivity and specificity. Chemiluminescence-based immunoassays have emerged as alternatives to ELISA methods because of their short testing duration, with some assays providing results in as little as 30 min. The validation process for these newer methods needs to be expanded to establish their performance equivalency with established ELISA systems [8,9]. In addition, the International Council for Standardization in Hematology (ICSH) has published best practice recommendations for the laboratory measurement of ADAMTS13 activity and for antigen and inhibitor assays. This expert panel recommendation emphasizes the importance of standardized ADAMTS13 testing to support better diagnosis and timely treatment, especially in patients with thrombotic microangiopathies [10].

There is also extensive evidence supporting the critical role of routine clinical ADAMTS13 testing in thrombotic thrombocytopenic purpura (TTP) patient diagnostics and prognosis evaluation. ADAMTS13 activity is significantly reduced or even absent in severe cases of TTP [11,12]. An ADAMTS13 level below 10% is generally considered diagnostic for TTP in clinical practice [1]. However, expert panels and guidelines now suggest including TTP in the differential diagnosis even with higher ADAMTS13 levels, owing to individual and laboratory testing variability. The International Society on Thrombosis and Hemostasis (ISTH) has integrated ADAMTS13 testing into its TTP diagnostic criteria [13]. In addition to TTP, ADAMTS13 measurement is essential in diagnosing other thrombotic microangiopathies, such as hemolytic uremic syndrome (HUS). Unlike TTP, the pathophysiology of HUS is generally triggered by bacterial toxins, and the latter disease predominantly affects younger patients. Various expert societies endorse ADAMTS13 testing in HUS for accurate diagnosis and management [13,14,15].

### Objectives and Research Gap

Although ADAMTS13 testing has become a key diagnostic tool in thrombotic microangiopathies, its broader clinical relevance is not yet well defined [16]. A proof-of-principle study by Reinecke et al. provided early evidence of a perioperative decrease in ADAMTS13 activity and an increase in large VWF multimers in cardiac surgery patients [17]. The scientific community has shown increasing interest in how ADAMTS13 activity changes during major surgical procedures, including cardiovascular interventions. ADAMTS13 was recently included in a biomarker panel in a study protocol evaluating endovascular aneurysm repair due to its potential as a candidate for detecting coagulation disturbances and endothelial activation in vulnerable patients [18]. Thus, a research gap regarding the suitability of ADAMTS13 in the perioperative setting remains, and there is a need to expand these preliminary observations to larger patient cohorts, accompanied by comprehensive analyses addressing biochemical and clinical parameters. To bridge this gap, we systematically assessed the perioperative dynamics of ADAMTS13 activity and VWF alterations in a substantially larger patient cohort than that reported by Reinencke et al. In addition, we incorporated data on clinical complications (e.g., postoperative new-onset bleeding events and thromboembolic events) and liver function markers to elucidate the underlying mechanisms.

## 2. Materials and Methods

Blood samples from 168 cardiac surgery patients were collected prospectively throughout their perioperative period. The study included patients who received cardiac surgery and excluded patients who were younger than 18 years old or pregnant women, as well as patients with preanalytical sample problems (coagulated samples, insufficient volume) or ADAMTS13 inhibitor levels beyond 15 U/mL. The initial screening of 171 patients led to the exclusion of 3 patients, thus making 168 patients eligible for the final analysis. The patient recruitment process is presented in Figure 1. A total of 628 ADAMTS13 activity measurements and parallel ADAMTS13 inhibitor measurements were performed. An initial assessment was performed to determine ADAMTS13 activity at baseline before anesthesia. Blood samples were collected at predefined perioperative timepoints: (1) baseline sample before induction of anesthesia, and (2) postoperative samples once daily on each of the first six days after surgery. All blood draws were performed using a single-puncture approach where possible, or via an existing central venous catheter or arterial line when available, according to clinical routine. The timing of blood draws was standardized and documented in the study protocol. Samples were drawn into 3.2% sodium citrate tubes and immediately processed for plasma separation by centrifugation.

The collected clinical data consisted of patient age and sex, together with surgical procedure type and essential laboratory markers, including liver function parameters (albumin, transaminases). The study population included patients undergoing coronary artery bypass grafting (*n* = 89), isolated heart valve replacement or repair (*n* = 57), and combined procedures (*n* = 22). These procedure types were chosen to represent the most common major cardiac surgeries performed with the use of heart–lung machines. The documentation of postoperative complications included bleeding, thromboembolic events, and organ dysfunction based on routine clinical assessments. Data extraction was performed from electronic medical records through standard perioperative documentation processes.

The Technozym ADAMTS13 activity chromogenic ELISA (Technoclone, Vienna, Austria) was used to measure ADAMTS13 activity following the manufacturer’s instructions. The reconstitution of the VWF73 substrate required 6 μL of distilled water and 15 min of incubation at room temperature (20–25 °C). One hundred microliters of substrate was added to wells coated with anti-glutathione S-transferase and incubated at room temperature for 60 min, followed by 3 wash cycles. Calibrators and control plasma samples were reconstituted in 500 μL of distilled water, thoroughly mixed, and incubated for 15 min. One hundred microliters of sample, control, or calibrator material was added to each well and incubated at room temperature for 30 min. Next, 100 μL of horseradish peroxidase conjugate was added to each well, followed by 100 μL of Tetramethylbenzidin (TMB) substrate. The mixture was then incubated for 30 min at room temperature. One hundred microliters of stopping solution was added to the wells, and the ELISA plate was measured at 450 and 620 nanometers (nm). The reference range for the Technozym ADAMTS13 activity assay is 0.4–1.3 IU/mL. The ADAMTS13 inhibitor was measured using the Technozym ADAMTS-13 INH ELISA which is a chromogenic enzyme immunoassay. The assay has a measuring range of 2–104 U/mL and a detection limit of 1.68 U/mL. The results are quantified by a standard curve calibrated against high-titer reference plasma (100 U/mL). According to the manufacturer, values >15 U/mL are considered positive. The Technozym ELISA was used to measure VWF antigen (VWF:Ag) and VWF collagen binding (VWF:CB), as per the manufacturer’s instructions. The VWF:CB assay was used to assess the A1 domain of VWF to determine its collagen-binding capacity, since this domain enables platelet adhesion during shear stress conditions. The VWF:CB/VWF:Ag ratio was calculated to assess acute-phase effects that change the absolute VWF levels by determining the quantity of functional high-molecular-weight VWF multimers relative to total VWF antigen levels. The relationship between VWF processing and ADAMTS13 activity can be monitored through this ratio since increased values indicate reduced VWF multimer cleavage by ADAMTS13.

Descriptive metrics were used to present the study variables together with their mean, standard deviation, and median values. An independent samples *t* test was used to examine preoperative–postoperative value differences when the data followed a normal distribution, and the Mann–Whitney U test was applied for nonnormally distributed data. Shapiro–Wilk testing at α = 0.05 and graphical assessments (histograms and QQ plots) were used to determine normality. Regression analyses were performed to investigate the relationships between ADAMTS13 activity and clinical factors, including VWF dynamics and liver function biomarkers, such as Cholinesterase (CHE), Albumin (ALB), Glutamat–Oxalacetat–Transaminase (GOT), and Glutamat–Pyruvate–Transaminase (GPT). Spearman’s correlation coefficients were calculated to assess relationships between these continuous variables. SPSS version 0.18.3 and R version 4.3.2 were used for statistical analysis. R was used to create all graphical illustrations and plots.

The study received Institutional Review Board approval from Upper Austria/Johannes Kepler University, with the approval code 1120/2023 on 5 September 2023, and was conducted in accordance with the principles of the 1964 Helsinki Declaration, along with its subsequent modifications and equivalent ethical standards.

## 3. Results

A total of 171 patients who underwent cardiothoracic procedures (heart valve replacement and/or aorto-coronary bypass surgery with HLM) were recruited, 168 of whom were included in the final analysis after excluding 3 patients whose ADAMTS13 inhibitor levels exceeded 15 U/mL. In total, 628 laboratory measurements were analyzed. The perioperative time frame led to a significant reduction in ADAMTS13 activity, which decreased from 0.83 IU/mL preoperatively to 0.65 IU/mL at the final postoperative measurement (*p* = 0.01) (Figure 2). In addition, a comparison between ADAMTS13 activity on the day of surgery (baseline) and on postoperative day 7 revealed a statistically significant decrease (*p* = 0.003).

The analysis of postoperative complications focused on thromboembolic events and new-onset bleeding complications. A trend toward a greater reduction in ADAMTS13 activity was observed in patients with complications, with values declining from 0.85 IU/mL preoperatively to 0.58 IU/mL at the last postoperative measurement compared with a decline from 0.82 IU/mL to 0.71 IU/mL in patients without complications (*p* = 0.13). The ADAMTS13 activity analysis results for patients with and without complications are presented in Figure 3.

We examined whether a decrease in ADAMTS13 activity results in decreased degradation of VWF molecules, thus leading to the accumulation of VWF molecules. In particular, we analyzed VWF:CB levels, which reflect the binding site of the alpha-1 domain of VWF and are indicative of high-molecular-weight VWF multimers. The ratio of VWF:CB to VWF:Ag was calculated to correct for acute phase variations occurring during the perioperative period. The analysis of VWF:CB/VWF:Ag ratio and ADAMTS13 activity showed an inverse relationship. Higher VWF:CB/VWF:Ag ratios were associated with lower ADAMTS13 activity levels. This opposing pattern was most evident in the cases with most extreme values. The highest VWF:CB/VWF:Ag ratio of 1.6 was observed with a markedly low ADAMTS13 activity of 0.66. The lowest ADAMTS13 activity value of 0.64 was associated with a relatively high ratio of 1.5. The relationship between ADAMTS13 activity and the VWF:CB/VWF:Ag ratio is presented in Figure 4 and Table 1.

Furthermore, we investigated how ADAMTS13 activity correlates with liver biomarkers, emphasizing choline esterase (CHE) as a key indicator of liver synthetic function. This analysis aimed to evaluate whether the observed decrease in ADAMTS13 activity was due to reduced ADAMTS13 production or increased ADAMTS13 consumption in the periphery. The results demonstrated that CHE was weakly correlated with ADAMTS13 activity (Spearman’s rho = 0.39), whereas other liver biomarkers, such as glutamate pyruvate transaminase (GPT) and glutamic oxalacetic transaminase (GOT), showed even weaker correlations. The regression analyses of ADAMTS13 activity correlation with CHE and liver biomarkers, including ALB, GPT, and GOT, are shown in Figure 5.

A summary of the main statistical results, including perioperative changes in ADAMTS13 activity and its correlations with clinical and biochemical parameters, is provided in Table 2.

## 4. Discussion

Perioperative complications remain the main issue in cardiac surgery, especially when patients are elderly or have multiple comorbidities. The main complications that occur include cardiovascular and cerebrovascular events, along with pneumonia, surgical site infections, acute kidney injury, bleeding or coagulopathy, and postoperative delirium. The severity and occurrence of these events depend on both the patient’s preoperative state and the intraoperative and postoperative care received. The current guidelines recommend using standardized tools like EuroSCORE II for preoperative risk assessment in addition to evaluating all relevant comorbidities, including renal and hepatic dysfunction and diabetes mellitus. The recommended approach for complication management requires surgical teams to work with anesthetic teams and intensive care and nursing teams for early identification and treatment of complications. The maintenance of stable hemodynamics across the perioperative period serves to reduce organ hypoperfusion risks. The prevention of postoperative delirium requires specific strategies, which include sedative reduction, early mobilization, sleep hygiene promotion, and catheter and restraint avoidance. The prevention of infections depends on prompt antibiotic prophylaxis administration following established protocols, together with strict hygiene practices. The management of antithrombotic and coagulation needs to be tailored to each patient to prevent both thrombosis and bleeding complications, especially when patients take anticoagulant medications [19].

The detection of ADAMTS13 deficiency remains essential for identifying thrombotic microangiopathies that produce life-threatening blood clots through impaired VWF cleavage [1,13,14]. ADAMTS13 is well-established for the diagnosis of TTP and HUS [2,15]. In addition, the established role of ADAMTS13 measurement in TTP diagnosis extends to its application in obstetric medicine for differentiating TTP from hemolysis, elevated liver enzymes, and low platelet count (HELLP) and atypical HUS. Therefore, the measurement of ADAMTS13 also serves as an exclusion biomarker to rule out TTP in pregnancy-related thrombotic microangiopathies [20]. Additional studies have expanded the clinical application of ADAMTS13 testing beyond thrombotic microangiopathies into liver diseases, as reduced ADAMTS13 levels directly correlate with the progression and severity of liver cirrhosis. Research has shown that patients with liver disease, especially those with autoimmune hepatitis, demonstrate decreased ADAMTS13 activity and elevated VWF levels; thus, ADAMTS13 is a reliable indicator of liver synthetic function [21,22]. ADAMTS13 testing applications are increasing as its evaluation expands in different clinical scenarios, including trauma, septic shock, sickle cell anemia, and multiple hematological disorders [23,24,25]. This prospective study presents a new use of ADAMTS13 testing for cardiac surgery patients. A significant reduction in ADAMTS13 activity occurred during this study: the preoperative level (0.83 IU/mL) decreased to 0.65 IU/mL after surgery (*p* = 0.01). Moreover, our data indicate that complications are associated with a greater decrease in ADAMTS13 activity (*p* = 0.13), which warrants further investigation. Prior medical research in cardiology demonstrated that elevated VWF in combination with decreased ADAMTS13 levels is associated with endothelial damage and the development of heart disease, including heart failure, atrial fibrillation, and myocardial infarction. Persistent imbalances between these elements are associated with adverse clinical outcomes and disease exacerbation [26]. In the context of cardiac surgery, Kokame et al. investigated nine patients with aortic stenosis who underwent aortic valve replacement in a small case series. They reported that the ratio of VWF antigen activity to ADAMTS13 activity decreased within one week; ADAMTS13 dynamics during the perioperative course were interpreted as adaptation mechanisms to new hemodynamic conditions. These promising results demonstrated that normalization of the VWF/ADAMTS13 ratio is an indicator of good surgical outcomes and recovery from hemostasis [27]. Although we also observed a significant decrease in ADAMTS13 activity, these changes appear to result from surgical stress and hemostatic activation effects rather than physiological changes.

The interaction of VWF with ADAMTS13, known as the VWF–ADAMTS13 axis, plays a vital role in cardiovascular injury. Gandhi et al. reported that ADAMTS13 acts as a protective factor against myocardial ischemia and reperfusion damage through its ability to cleave VWF, which prevents thrombosis, inflammation, and cell death [28]. Clinical studies, such as that by Eerenberg et al., highlighted that imbalances in the VWF–ADAMTS13 axis contribute to myocardial infarction patient outcomes. The combination of high VWF activity and low ADAMTS13 levels resulted in severe complications, including intramyocardial hemorrhage [29]. Our study supports these findings, revealing an inverse relationship between ADAMTS13 activity and the VWF:CB/VWF:Ag ratio, which is indicative of the accumulation of higher-molecular-weight VWF. The observed dynamic changes in ADAMTS13 activity and VWF indicate a fundamental interaction between these two molecules during the postoperative phase.

Correlation analyses revealed only a weak correlation between ADAMTS13 activity and parameters reflecting liver synthetic function, including liver biomarkers such as albumin, CHE, GPT, and GOT. Therefore, our results suggest that the postoperative reduction in ADAMTS13 activity is not caused by impaired liver function but primarily by mechanical stressors, such as shear stress that occurs during HLM and surgical procedures, which function independently of peripheral consumption. This identified mechanism of decreased postoperative ADAMTS13 activity stands out from well-documented concepts. In thrombotic microangiopathies, the primary cause of ADAMTS13 deficiency is autoantibodies that disrupt its function [30]. In sepsis patients, ADAMTS13 is depleted primarily through cytokine-mediated suppression. Previous studies have shown that elevated interleukin-6 levels, among other proinflammatory cytokines, directly correlate with decreased ADAMTS13 activity in patients with sepsis. The cytokine-mediated suppression of ADAMTS13 activity results in VWF imbalances, which lead to microvascular thrombosis and unfavorable clinical outcomes, particularly in patients with disseminated intravascular coagulation [31,32]. In contrast, our findings differ from previously described pathophysiological mechanisms establishing inflammation or immune system responses as the cause of ADAMTS13 reduction. The unique hemostatic difficulties experienced during cardiac surgery were apparent in our study. Our findings extend the current understanding of ADAMTS13 and support further investigations of surgical ADAMTS13 mechanisms.

Reinecke et al. investigated prospective changes in ADAMTS13 activity and VWF multimer concentrations in 47 cardiac surgery patients who presented decreased postoperative ADAMTS13 levels but did not develop thrombotic microangiopathy [17]. Our current study verifies these perioperative patterns in a larger population (*n* = 168) and extends this knowledge via the incorporation of VWF:CB/VWF:Ag ratios, liver function markers, and thromboembolic and bleeding events. The detailed laboratory evaluations and extended monitoring period in our study provide an additional understanding of how hemostasis changes during cardiac surgery. Reinecke et al.’s study was one of the first prospective studies to show that cardiac surgical procedures, including those using extracorporeal circulation, cause transient disturbances in the VWF–ADAMTS13 axis. However, Reinecke et al. did not find any correlation between the ADAMTS13 activity level and the accumulation of large VWF multimers [17]. Our data revealed a consistent inverse relationship between ADAMTS13 activity and the VWF:CB/VWF:Ag ratio. These results suggest that the reduction in ADAMTS13 activity could be associated with the accumulation of high-molecular-weight VWF multimers. We confirmed the main findings of Reinecke et al.’s study (significant decrease in ADAMTS13 from the pre- to postoperative timepoints) and addressed some of their analytical limitations. We implemented clinical outcome parameters, including thromboembolic and bleeding events. These complications occurred in patients with a more pronounced decrease in ADAMTS13 activity, potentially indicating a pathophysiological link.

The observations of this study demonstrate a wider interest in ADAMTS13 as a prognostic marker in different clinical settings outside of thrombotic microangiopathies and, therefore, follow a scientific trend of new clinical ADAMTS13 characterization and alternative use [4,33,34]. Our study broadens the knowledge about ADAMTS13 in the perioperative setting. However, whether these ADAMTS13 dynamics can predict long-term outcomes in cardiac surgery patients remains an important question for future research.

The study contains multiple limitations that need to be recognized. First, the study was conducted as a single-center observational study, which might affect the ability to generalize its results to different patient groups and operating environments. Second, the study recorded clinical complications, including bleeding and thromboembolic events, but these events were not established primary endpoints and were evaluated in an exploratory manner; therefore, the study was not powered to identify outcome variations or establish clinical predictive relationships. Lastly, the measurement of ADAMTS13 activity depended on a particular commercial assay; however, results may differ between different testing platforms.

## 5. Conclusions

Our study demonstrated that ADAMTS13 activity decreases significantly during cardiac surgery. Patients who developed postoperative complications such as thrombophilia or new-onset bleeding had lower postoperative ADAMTS13 levels than those without complications. This finding suggests that ADAMTS13 could serve as a perioperative marker for hemostatic imbalance; however, its predictive value, particularly in its use as a prognostic biomarker for long-term complications, needs additional validation.

## Figures and Tables

**Figure 1 jcm-14-04936-f001:**
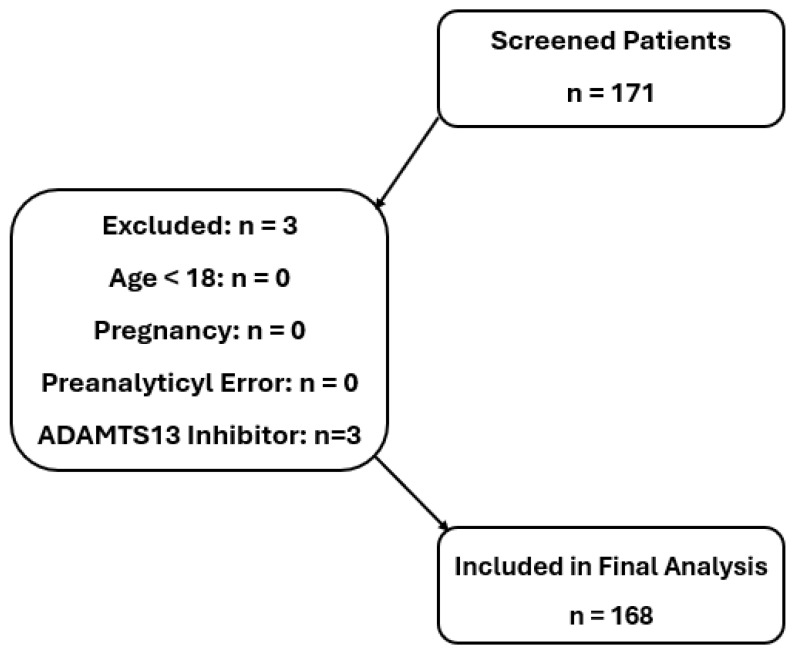
Patient recruitment flowchart. Of the 171 screened individuals undergoing cardiac surgery, 3 patients were excluded due to predefined exclusion criteria.

**Figure 2 jcm-14-04936-f002:**
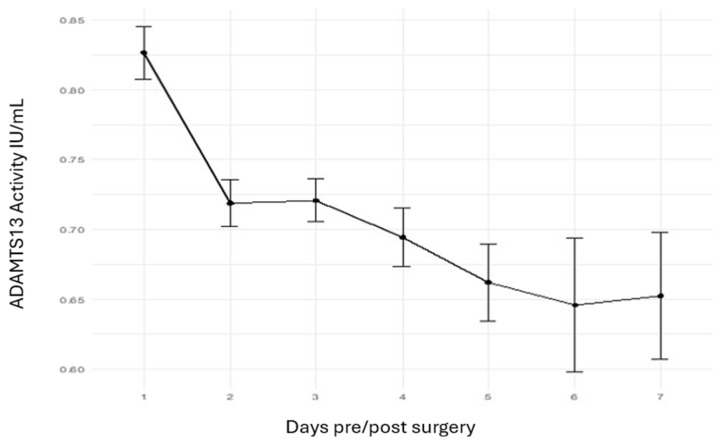
The time course of ADAMTS13 activity in patients undergoing cardiac surgery. The results revealed a significant reduction in ADAMTS13 levels after surgery compared with before surgery.

**Figure 3 jcm-14-04936-f003:**
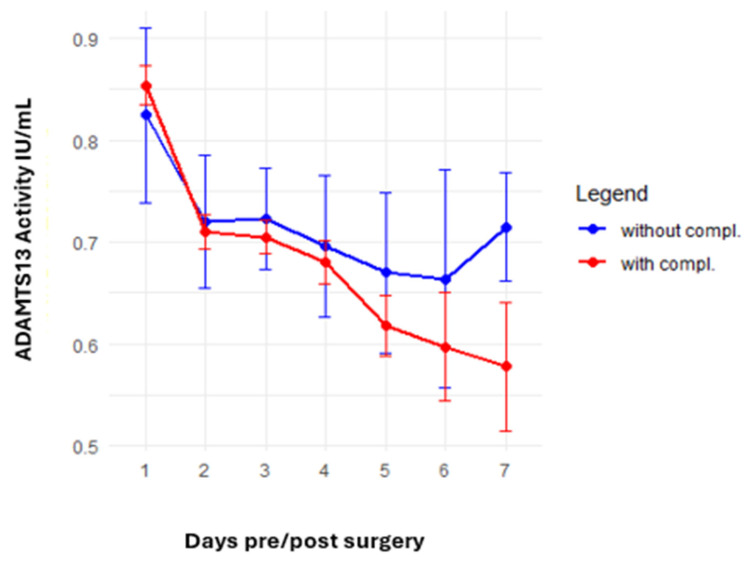
Comparison of ADAMTS13 activity between patients with and without thromboembolic or new-onset bleeding complications. A greater decrease in ADAMTS13 activity was noted in the complication group.

**Figure 4 jcm-14-04936-f004:**
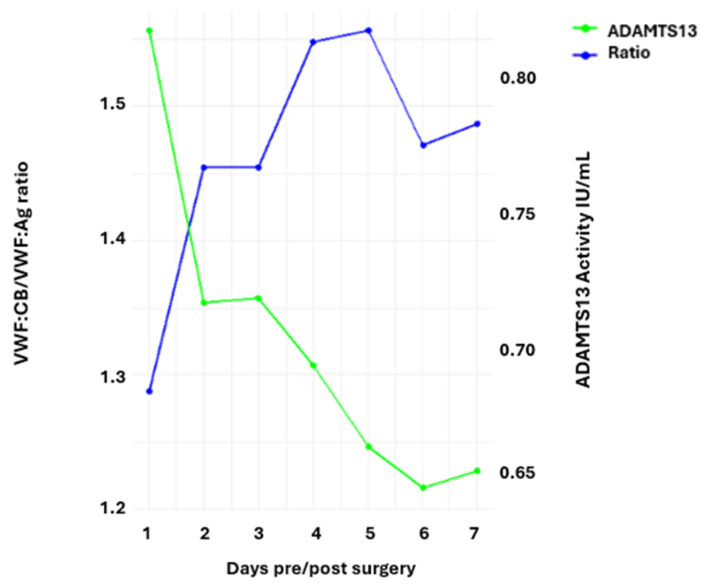
Inverse relationship between ADAMTS13 activity and the ratio of von Willebrand factor collagen binding (VWF:CB) to von Willebrand factor antigen (VWF:Ag).

**Figure 5 jcm-14-04936-f005:**
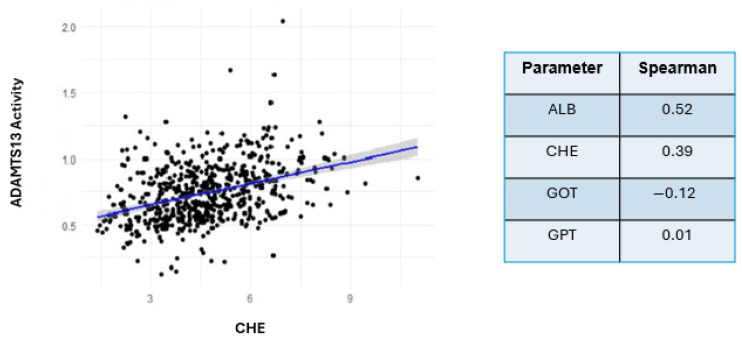
Relationships between ADAMTS13 activity and the liver biomarkers albumin (ALB), cholinesterase (CHE), glutamate–pyruvate transaminase (GPT), and glutamate–oxaloacetate transaminase (GOT). The weak associations between ADAMTS13 activity and liver function biomarkers suggest that the reduction in ADAMTS13 activity primarily arises from peripheral utilization rather than impaired hepatic production.

**Table 1 jcm-14-04936-t001:** Ratio of von Willebrand factor collagen binding (VWF:CB) to von Willebrand factor antigen (VWF:Ag) and the corresponding activity of ADAMTS13.

Measurement	VWF:CB/VWF:Ag Ratio	ADAMTS13 Activity
1	1.29	0.83
2	1.45	0.72
3	1.45	0.72
4	1.54	0.69
5	1.56	0.66
6	1.47	0.64
7	1.48	0.65

**Table 2 jcm-14-04936-t002:** Summary of statistical analyses associated with ADAMTS13 activity.

Analysis	Result
Pre- vs. postoperative ADAMTS13 activity	Significant decrease (0.83 → 0.65 IU/mL); *p* = 0.01
ADAMTS13 activity vs. VWF:CB/VWF:Ag ratio	Inverse correlation observed
ADAMTS13 activity vs. ALB	Spearman’s rho = 0.52 (moderate positive)
ADAMTS13 activity vs. CHE	Spearman’s rho = 0.39 (weak positive)
ADAMTS13 activity vs. GOT	Spearman’s rho = −0.12 (very weak negative)
ADAMTS13 activity vs. GPT	Spearman’s rho = 0.01 (no correlation)
ADAMTS13 activity in patients with complications	Greater decrease: 0.85 → 0.58 IU/mL
ADAMTS13 activity in patients without complications	Smaller decrease: 0.82 → 0.71 IU/mL; *p* = 0.13

## Data Availability

The original contributions presented in this study are included in the article. Further inquiries can be directed to the corresponding author.

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
