# Peer review of "Perioperative Profiling of a Disintegrin and Metalloprotease with Thrombospondin Type 1 Motif, Member 13 (ADAMTS13) Activity in Cardiac Surgery: Kinetics and Mechanistic Insights"

_jcm, 2025, doi:10.3390/jcm14144936_

Round 1

Reviewer 1 Report

Comments and Suggestions for Authors

The paper called “Perioperative Profiling of ADAMTS13 Activity in Cardiac Surgery: Kinetics, Mechanistic Insights, and Clinical Associations” offers a new look at how ADAMTS13 activity changes during heart surgery, building on earlier research by including liver function markers and a bigger group of patients. Overall, the study is scientifically sound, and the data support the conclusions drawn. However, the English writing could be polished further to ensure better clarity and precision in conveying the scientific message. Minor edits for grammar and style would enhance readability.

Author Response

Reviewer 1:

The paper called “Perioperative Profiling of ADAMTS13 Activity in Cardiac Surgery: Kinetics, Mechanistic Insights, and Clinical Associations” offers a new look at how ADAMTS13 activity changes during heart surgery, building on earlier research by including liver function markers and a bigger group of patients. Overall, the study is scientifically sound, and the data support the conclusions drawn. However, the English writing could be polished further to ensure better clarity and precision in conveying the scientific message. Minor edits for grammar and style would enhance readability.

Reply: We appreciate the recognition of the scientific merit. We have thoroughly revised the manuscript to improve English grammar, spelling, and overall clarity.

Reviewer 2 Report

Comments and Suggestions for Authors

While the manuscript and its subject are very interesting, there are several issues that need addressing:

  1. The most poignant is the number of abbreviations used first time withouts explanation such as ADAMTS13 and vWF in the abstract.
  2. The title is proper, yet it uses another unexplained abbreviation
  3. The abstract is way too simple, especially with regard to the background and conclusions which could use further developments.
  4. The first paragraphs in the introduction are written very abrupt. There should be a better linearity in reading them. Also, avoid several repetitions.
  5. In the material and methods section, the inclusion and exclusion criteria could be mentioned along with the number of subjects who were considered but did not fulfill certain criteria. This could be achieved through a diagram or figure.
  6. The study approval number should also be mentioned in the text.
  7. The results could use some tables showing the statistical analysis values. Also, in figure 4, the tables shows Spearmen instead of Spearman.
  8. The authors should also be more explicit regarding the heart procedures the patients underwent and why did they choose those in particular.
  9. Also, either in the introduction or discussion, there should be an emphasis on the most frequent post-operatory complications and current guidelines for screening or prevention.
  10. The conclusion properly summarize the findings of the study.
  11. The limitations of the study should be presented, perhaps at the end of the discussions.

Author Response

Reply to the reviewer

Reviewer 2:

While the manuscript and its subject are very interesting, there are several issues that need addressing:

  1. The most poignant is the number of abbreviations used first time withouts explanation such as ADAMTS13 and vWF in the abstract.

Reply: We have revised the abstract to include definitions for all abbreviations at first mention

  1. The title is proper, yet it uses another unexplained abbreviation.

Reply: The abbreviation ADAMTS13 has now been fully spelled out upon first mention in the title.

  1. The abstract is way too simple, especially with regard to the background and conclusions which could use further developments.

Reply: We have expanded the background section of the abstract for better context and the conclusion section  to more specifically reflect our main findings and their clinical implications.

  1. The first paragraphs in the introduction are written very abrupt. There should be a better linearity in reading them. Also, avoid several repetitions.

Reply: We have revised the introduction and carefully edited the entire manuscript to improve readability.

  1. In the material and methods section, the inclusion and exclusion criteria could be mentioned along with the number of subjects who were considered but did not fulfill certain criteria. This could be achieved through a diagram or figure.

Reply: We have now added the specific inclusion and exclusion criteria to the Materials and Methods section. Additionally, we included a flowchart to illustrate the recruitment process

  1. The study approval number should also be mentioned in the text.

Reply: We mentioned the approval code.

  1. The results could use some tables showing the statistical analysis values. Also, in figure 4, the tables shows Spearmen instead of Spearman.

Reply: We corrected Spearman analysis and added a results summary table to the manuscript.

  1. The authors should also be more explicit regarding the heart procedures the patients underwent and why did they choose those in particular.

Reply: We have now added more detailed information to the Methods section to clarify the types of cardiac procedures performed. Specifically, our study cohort included patients who underwent coronary artery bypass grafting (n = 89), isolated valve surgery (n = 57), and combined interventions (n = 22). These procedures were selected as they represent the most frequent and clinically relevant forms of cardiac surgery involving extracorporeal circulation.

  1. Also, either in the introduction or discussion, there should be an emphasis on the most frequent post-operatory complications and current guidelines for screening or prevention.

Reply: We have incorporated a summary of the most common postoperative complications following cardiac surgery into the discussion section, along with guideline-based strategies for perioperative prevention.

  1. The conclusion properly summarize the findings of the study.

Reply: Thank you for this positive remark.

  1. The limitations of the study should be presented, perhaps at the end of the discussions.

Reply: We have now included a paragraph at the end of the Discussion section to explicitly address the limitations of our study.

Reviewer 3 Report

Comments and Suggestions for Authors

Dear authors,

thanks for conducting the present investigation, in which you tried to draw a relation between ADAMTS13, vWF and clinical aspects in the perioperative period around heart surgery. 

Although the topic is interesting, the manuscript misses relevant data of the clinical aspects. You do not describe which data were collected, neither do you present some. The method of how you recorded this information is fully unclear. In that sense, I propose to leave the clinical aspects behind and focus on the laboratory part, only. 

The introduction reads more like an advertisement for the assay. So, I strongly suggest to reduce mentioning the manufactures to a minimum. 

The result section gives three tables in which the levels of aDAMTS13 are different (in particular 1 vs 3). The relation with clinical data is weak and the liver enzyme comparison does not take into account a potential rise due to cardiac ischemia.

The discussion is weak and does not give a good explanation why the levels are decreasing. 

Please find more details in the attached pdf. 

Comments on the Quality of English Language

Although generally the language is acceptable, a native speaker review will improve readability. 

Author Response

Reply to the reviewer

Reviewer 3:

Dear authors,

thanks for conducting the present investigation, in which you tried to draw a relation between ADAMTS13, vWF and clinical aspects in the perioperative period around heart surgery. 

Although the topic is interesting, the manuscript misses relevant data of the clinical aspects. You do not describe which data were collected, neither do you present some. The method of how you recorded this information is fully unclear. In that sense, I propose to leave the clinical aspects behind and focus on the laboratory part, only. 

Reply: First of all, we thank the reviewer for this valuable comment and for considering our topic to be of interest. We agree that a more precise description of the clinical data collection was necessary and have therefore revised the Methods section accordingly. It now clearly outlines which clinical parameters were collected (e.g., age, sex, type of surgery, liver biomarkers, and postoperative complications) and specifies that data were extracted from electronic medical records as part of routine clinical documentation and as a prospective observational study.

The introduction reads more like an advertisement for the assay. So, I strongly suggest to reduce mentioning the manufactures to a minimum. 

Reply: We have revised the Introduction to reduce emphasis on manufacturer names and focused the text more on the scientific rationale of the study. However, we would like to note that in laboratory-based clinical research, it is important to specify manufacturers and assay details in the Methods section to ensure reproducibility and comparability of results.

The result section gives three tables in which the levels of aDAMTS13 are different (in particular 1 vs 3). The relation with clinical data is weak and the liver enzyme comparison does not take into account a potential rise due to cardiac ischemia.

Reply: Adaptions were made in the legends and text to improve clarity.

The discussion is weak and does not give a good explanation why the levels are decreasing. 

Reply: We would like to emphasize that, to date, only a single proof-of-concept study in a smaller subset of patients exists. This study was already cited in both the Introduction and Discussion of this manuscript. Consequently, the available literature and high-level evidence on this specific topic remain extremely limited. Given this gap, we structured our Discussion to carefully integrate our clinical findings with mechanistic insights from preclinical and observational studies, particularly those related to von Willebrand factor biology, inflammation, and endothelial injury. In the revision, we have also included additional considerations on aortic stenosis and von Willebrand factor alterations, as these are clinically relevant and mechanistically linked to ADAMTS13 activity.

Please find more details in the attached pdf. 

Reply:

We have carefully reviewed all suggestions of the attached pdf.

In particular, we revised the introduction by shortening the technical sections related to the assay. While we reduced overly technical or manufacturer-related content, we deliberately retained the brief mention of thrombotic thrombocytopenic purpura and hemolytic uremic syndrome in the introduction. These represent the standard clinical indications for ADAMTS13 testing and were included to provide essential background, helping to contextualize the novelty of applying this biomarker in a perioperative cardiovascular setting. This also serves to introduce the rationale for the present study and to frame the study objectives. A major point raised in the reviewer’s comments concerned the clinical integration of the findings. We agree that a clearer distinction between the type and purpose of the study was needed. This is a clinical, observational study with a focus on laboratory medicine. Therefore, we revised the manuscript to emphasize that the core objective was to describe the kinetics of ADAMTS13 in the perioperative course and to explore its dynamic relationship with von Willebrand factor. While clinical complications such as thrombosis or bleeding were documented, these were considered secondary observations. The role of clinical outcome has thus been de-emphasized accordingly in the revised text. In line with this, we also adjusted the wording throughout the manuscript. References to broader clinical associations or outcome analysis have been replaced with a more specific and appropriate focus on postoperative complications. To reflect this conceptual clarification, we further modified the manuscript title, removing any implication of a primary outcome-driven analysis and focusing instead on the perioperative dynamics of ADAMTS13.

Round 2

Reviewer 2 Report

Comments and Suggestions for Authors

I consider all my observations have been properly addressed and that the manuscript can be published in its current form.

Author Response

Thank you for reviewing the manuscript.